# The development of numerical mapping in preschool children

**Jun Zhu[1,2], Huanyu Yang[3], Liangzhi Jia[1], Chenli Li[1], Qiang Wu[1], Yajie Bi[1], Fangwen Yu[1], Yun Pan [1,4]***

**1** School of Psychology, Guizhou Normal University, Guiyang, China, **2** Xichang University, Xichang, China, **3** Yunnan Minzu University, Kunming, China, **4** Guizhou Education University, Guiyang, China

* panyun129@163.com

## Abstract

The developmental order of numerical mapping and the symmetry of the mapping direction in preschool children remain controversial. In light of this context, this study investigated the developmental characteristics of numerical mapping among preschool children in China. In Experiment 1, 128 children aged 2–4 years were recruited to participate in six numerical mapping tasks with small and large numbers. The results showed no reliable differences between the mapping directions. Notably, the developmental orders of small- and large-number mapping differed, potentially confirming two systems of number representation. Furthermore, previous studies have found that counting or not could affect children's performance on numerical tasks. Therefore, in Experiment 2, we introduced the condition of "reminding children to count" and recruited 97 children aged 3–4 years to explore its influence on numerical mapping. Our findings revealed that reminding children to count could improve their performance on numerical mappings, resulting in alterations in the symmetry of the mapping direction and developmental order of mapping paths. In conclusion, number size and counting reminders would affect researchers' assessments of the developmental order of numerical mapping in preschool children, which serves as a guide for nurturing and evaluating the numerical mapping ability of preschool children.

## Introduction

People typically transmit numerical information in one non-symbolic format (arrays) and two symbolic formats (number words and Arabic digits). Dehaene (1992) proposed a triple-code model, suggesting that numbers are mentally processed in analogical code (matching arrays), auditory verbal code (matching number words), and visual Arabic number code (matching digits) [1–3]. Subsequently, researchers began to investigate mutual mapping among these three codes and the developmental characteristics of numerical mapping in early childhood [4–12]. Mutual mapping

**Data availability statement:** To facilitate access to our data, we have uploaded the stimuli, raw data, and related experimental materials to the Open Science Framework at https://osf.io/rs2up/.

**Funding:** This work was supported by the National Natural Science Foundation of China (grant number 32360203); and the Major project of Key Research base of Humanities and Social Sciences of Ministry of Education (grant number 22JJD190009). The funders had no role in study design, data collection and analysis, decision to publish, or preparation of the manuscript.

**Competing interests:** The authors have declared that no competing interests exist.

among the three codes generates three mapping paths: digit/number word (path 1), array/number word (path 2), and digit/array (path 3) [8].

## Mapping direction

Before 2013, researchers mainly focused on one or two mapping paths [13–16]. Benoit et al. (2013) extended this work by investigating the development of all three mapping paths in French children [5]. They assessed 3- to 5-year-old children's ability to map between arrays, number words, and digits for small (1–3) and large numbers (4–6), using six-alternative forced-choice or direct quantity naming tasks without providing counting reminders. Regarding mapping direction, their results revealed no significant differences in children's performance between directions, such as array-to-number word (AN) and number word-to-array (NA). They proposed two explanations for why children might acquire both mapping directions simultaneously. First, children are likely exposed to both mapping directions with comparable frequency in similar environments. Second, children may connect digits to an abstract understanding of number, which they develop when initially forming the mapping between number words and dot arrays.

Jiménez Lira et al. (2017) examined 2- to 4-year-old Canadian children's numerical mapping abilities using two-alternative forced-choice or direct quantity naming tasks, without providing explicit counting instructions, across small (1–3), medium (4–6), and large (7–9) number ranges. However, they found significant differences between the mapping directions of paths 2 and 3 [8]. In path 2, children performed better on number word-to-array (NA) mapping than on array-to-number word (AN) mapping [17], and similarly, in path 3, children were more accurate on digit-to-array (DA) mapping than on array-to-digit (AD) mapping for small set sizes. Since both paths involved arrays, the directional differences may stem from the fact that selecting between two arrays (NA or DA) is cognitively easier than producing an exact number from an array (AN or AD).

In contrast, Hurst et al. (2017) investigated 3- and 4-year-old American children's number mappings using five-alternative forced-choice tasks, with number words also provided as candidate options, across small (1–3) and large (4–5) number ranges. They observed that children performed slightly better on AN than on NA mapping [7]. The researchers proposed that the observed performance difference might be attributed to the use of counting. In the NA task, children were presented with five distinct sets of dots, which may have been overwhelming and thus reduced their likelihood of engaging in counting, whereas the AN task involved only a single set of dots, making spontaneous counting more likely.

Hurst et al. (2017) suggested that although slight directional differences may emerge under specific conditions (such as in path 2), these differences are likely attributable to the counting demands of the task rather than indicating a fundamental cognitive advantage in one mapping direction [7]. Consistent with this, Benoit et al. (2013) also found that children demonstrated bidirectional symmetry in numerical mapping tasks [5]. Therefore, the first aim of this study was to examine the directional symmetry across the three numerical mapping paths, with the hypothesis that mapping direction would be symmetrical.



## Developmental order of numerical mapping

Benoit et al. (2013) found that children first learned to map in path 2, then path 3, and finally path 1 [5]. Therefore, the developmental order of numerical mapping was reported to be 2→3→1. Bialystok (1992) outlined three stages in children's development of symbolic representation [18]. Initially, children can recite a sequence of number words by which they learn to count the arrays (path 2) [19]. Then, they learn to map in path 1, before finally, learning to map in path 3 (2→1→3). Knudsen et al. (2015) suggested that children first acquire mapping path 2 and then acquire mapping paths 1 and 3 simultaneously (2→1 and 3) [20]. Jiménez Lira et al. (2017) found that more children successfully mapped in path 1 than 2, and children were least likely to successfully map in path 3 (1→2→3) [8]. Both Hurst et al. (2017) and Marinova et al. (2021) demonstrated that children first acquire mapping paths 2 and 1, and only acquire mapping path 3 afterwards (2 and 1→3) [7,9]. Furthermore, they found that mapping path 1 significantly mediated the relationship between paths 2 and 3.

Benoit et al. (2013) proposed two possible explanations in the *Introduction* section for why children tend to acquire mapping path 1 before mapping path 3. [5]. First, children often recite the number word list as a jingle, which provides them with more practice in mapping path 1. Second, they can utilize one-to-one correspondence (one number word/ one digit) to easily map in path 1. However, children may not be able to easily map in path 3 because they utilize one-to-many or many-to-one correspondence (one digit/ many dots in an array). In the *Discussion* section, the authors also offered two alternative explanations for why children acquire path 3 before path 1. First, they suggested that non-symbolic number systems serve as the foundation for symbolic number systems. It is likely easier for children to map between symbolic and non-symbolic codes (path 3) than to map directly between two symbolic codes (path 1). Second, arrays and digits were perceptually available throughout the task, whereas number words were only fleetingly present. Therefore, it may be easier for children to map in path 3 than in path 1.

Hurst et al. (2017) proposed two main accounts in the *Introduction* section for the developmental order of numerical mapping: the "quantity account" and the "symbolic account" [7]. The "quantity account" postulates that children acquire direct mapping path 3 just as they had previously learned to map in path 2. Only after acquiring these two mapping paths can children learn to map between two symbolic formats (path 1) based on corresponding non-symbolic arrays [5,9]. The "quantity account" is consistent with the four-step developmental model [21] which considers the analogical code to have a crucial role in verbal and digital codes [2,22]. It assumes that the innate analogical code provides the basic meaning of numbers (step 1), which is the foundation of association arrays with number words (step 2) and later with digits (step 3). In contrast, the "symbolic account" posits that children initially acquire mapping path 2, then acquire path 1, and finally, associate digits with corresponding arrays (path 3) based on paths 1 and 2 [7,9]. In this view, mapping path 1 is a mediator between paths 2 and 3. Ultimately, the results of Hurst et al.'s study were consistent with the symbolic account, suggesting that symbolic knowledge plays a central role in the development of numerical mapping skills.

In light of the conflicting findings and theoretical interpretations regarding the developmental order of numerical mapping, the present study aimed to systematically examine the order in which children acquire the three numerical mapping paths, and to evaluate the explanatory power of the "quantity account" and the "symbolic account" in shaping this developmental trajectory.

## Mapping differences by number size

Previous studies have found that children learn numerical mapping for small numbers before large numbers [7,8,23]. Benoit et al. (2013) found that the mapping performance on path 2 of children aged 3 was better for small numbers (1–3) than for large numbers (4–6) [5]. Meanwhile, 4-year-old children performed better for small numbers than for large numbers on the three mapping paths. Hurst et al. (2017) demonstrated that children aged 3–4 years performed better for small numbers (1–3) than for large numbers (4–5) on paths 1 and 3 [7]. Jiménez Lira et al. (2017) discovered the effects of number size for both paths 1 and 2 [8]. Lipton and Spelke (2005) divided 5-year-old children into skilled and unskilled counters

and found that unskilled counters performed better in mapping number words to arrays for smaller numbers than for larger numbers beyond their counting range [13].

Also, children typically start with smaller numbers when acquiring numerical information. Around 2 years old, children begin to understand the cardinal meanings of the numbers "one", "two", and "three" sequentially. Around age 3, children learn verbal counting and begin to understand the basic concepts of symbolic numbers [24–28]. They can gradually use counting to determine the cardinality of an array based on the "last word rule" [27,29–31]. After learning the cardinal meanings of numbers up to approximately "four", children appear to make rapid progress in understanding larger numbers [27,28,32,33].

Therefore, researchers have suggested the processes for learning small and large numbers require distinct cognitive mechanisms [33–40]. For example, an object tracking system (OTS) was considered to accurately represent small numbers (1–3 or 4) [36,41,42], and an approximate number system (ANS) was considered to approximately represent larger numbers (> 4) [36,43–46]. Specifically, for a small number range, people can obtain the accurate number of an array quickly through "subitizing" [47] and obtain the approximate number by estimation in a limited time for non-symbolic numbers greater than four [48,49]. Starkey & Cooper Jr (1995) showed that the subitizing range increased with age during early childhood from 1–3 to 1–5 (the subitizing range of adults) [50].

Building on prior research showing that children tend to acquire numerical knowledge for small numbers earlier than for larger ones, this study aimed to examine whether the developmental order of numerical mapping differs between small (1–3) and large (4–6) numbers. Given that small numbers are often processed through OTS and large numbers through ANS, we hypothesized that the developmental trajectories of mapping paths may vary across these number ranges, potentially reflecting differences in underlying cognitive mechanisms.

## Effect of counting on numerical mapping

Previous research showed that children's performance on numerical tasks improved when they spontaneously used counting strategies [51–57], and counting proficiency was closely linked to numerical mapping, especially in populations like deaf children [58]. For example, Walker et al. (2024) compared deaf and hearing children and found that counting skills were crucial for the development of age-typical numerical mapping abilities. Chan et al. (2020) demonstrated that counting helped kindergarteners form mappings between symbolic number words and non-symbolic quantities [51]. Posid and Cordes (2015) found that spontaneous counting mitigated biases in abstract numerical judgments affected by perceptual variability [53], while Posid and Cordes (2018) showed that brief counting instruction significantly improved children's understanding of cardinality, even increasing the likelihood of spontaneous counting during tasks [54]. These findings emphasize the critical role of counting in early numerical mapping development.

Among various numerical mapping tasks, the Give-N task specifically involves mapping symbolic numbers to non-symbolic quantities and has been widely used to assess children's numerical concept development [27,32,59–61]. Originally introduced by Wynn (1990) [27], this task asks children to give a specific number of items (e.g., "Can you give me three blocks?"). Based on their responses, children are typically categorized as "subset-knowers" (who can reliably provide small quantities such as 1, 2, or 3) or "cardinality-principle-knowers" (who can correctly provide larger quantities and demonstrate an understanding that the final number word in a count represents the total set size). The Give-N task has since become a widely adopted and robust method in developmental psychology for evaluating children's understanding of number concepts.

However, some researchers have questioned the reliability of the Give-N task, arguing that it may underestimate children's number knowledge when counting is not explicitly prompted [24,62,63]. Spontaneous counting does not consistently occur across task conditions [7,64,65], and performance can be influenced by follow-up questions. For example, Krajcsi (2021) found that asking children to recount the items improved their accuracy and led to higher knower-level classifications [62]. Baroody et al. (2023) further noted that the traditional Give-N task may underestimate young children's

understanding of small numbers, as it requires checking an initial response by one-to-one counting, confounds pre-counting and counting competencies [24]. These findings highlight the importance of considering the role of counting in both the design and interpretation of Give-N tasks.

In light of these concerns, we introduced a "counting reminder" condition to examine its effect on assessing numerical mapping development, and hypothesized that reminding children to count would improve their performance on numerical mapping tasks.

## The current study

In summary, the developmental order of numerical mapping and the symmetry of the mapping direction in preschool children remain controversial. The reasons may include variations in experimental materials (especially number size), inconsistent experimental procedures, and participants from different socioeconomic backgrounds or education levels [5,7–10,66–68]. Studies have shown that children from lower socioeconomic backgrounds tend to perform worse on non-symbolic number processing and numerical mapping tasks, which is closely linked to lower overall math achievement [66–68]. In this study, we primarily focused on number size and whether children were reminded to count during the experimental procedure to explore the developmental characteristics of children's numerical mapping, while controlling for the influence of socioeconomic background.

We hypothesized that (1) the mapping directions have no reliable differences between them, because our experimental materials are similar to Benoit et al. (2013) [5], (2) the developmental order of numerical mapping paths would differ between small and large numbers—for example, under the condition without counting reminders, we expect the best performance on path 2 (array/number word) for small numbers, and the best performance on path 1 (digit/number word) for large numbers; and (3) reminding children to count would improve their performance on paths 2 and 3 for large numbers, which require counting, thereby altering the symmetry of mapping directions and the developmental order of mapping paths.

## Experiment 1

### Method

**Participants.**  Experiment 1 initially included 128 children. After excluding six invalid participants, including two over 4 years of age and four (1 two-year-old and 3 three-year-olds) who did not undergo the entire procedure, 122 remained. To make our sample as comparable to that of Benoit et al. (2013) we further divided the children into three age groups: 40 two-year-olds (20 male; range = 24–35 months; M = 2 years 7 months), 45 three-year-olds (23 male; range = 36–47 months; M = 3 years 7 months), and 37 four-year-olds (20 male; range = 48–59 months; M = 4 years 4 months). Children were recruited from five preschools in China: three in Dangwu Town, Guiyang City, Guizhou Province, and two in Mengyang Town, Ya'an City, Sichuan Province. A stratified random sampling method was employed within each preschool, ensuring equal representation of boys and girls across the selected age groups. The experiment was conducted from April 17th to May 23rd in 2023. This study was approved by the Human Subjects Protection Committee of the School of Psychology at Guizhou Normal University (Approval No. GZNUPSY.N 202303 E [0004]). Permission to conduct the test was obtained from the legal guardians of the participants with school leaders present as a witness to ensure that the procedure was carried out appropriately. The children voluntarily participated and were allowed to stop the test at any time. They each received a gift for their participation.

**Stimuli and materials.**  The experimental materials were presented in a printed format for all participants, as printed materials offer a more developmentally appropriate and child-friendly mode of interaction for young children. Compared to screen-based formats, printed materials allow for more intuitive engagement and reduce potential distractions related to digital interfaces. The design and spatial layout of the stimuli (including dot arrays and digits) were adapted from Benoit et al. (2013) [5], who used laptop-based presentations. While the mode of presentation differed, we maintained the same

core features in stimulus design to ensure comparability. This approach also aligns with previous research using printed stimuli with preschool-aged children [8]. Red dot arrays (pattern on dice) and digits (Times New Roman, in bold) were printed on standard A4 copper plate paper (128 g/page). Each page included upper and lower black rectangular borders (19 x 13.5 cm) with a smaller black rectangle (10.5 x 7.5 cm) inside to display the stimuli. Small (1, 2, 3) and large (4, 5, 6) numbers were used in each mapping task, with each number having two trials for 72 trials in total [5]. The sequence of the target numbers was fixed according to the first two lines of a Latin square design, which included small (1, 2, 3, 2, 3, 1) and large numbers (4, 5, 6, 5, 4, 6). In contrast, the order of the six alternative numbers (arrays and digits) in the last four mapping tasks was randomly generated (Fig 1).

**Tasks.  Digit-to-Number Word Task:** The children were shown one digit (no.150 font) per trial in a rectangle at the bottom of the paper. They were asked to say the digit aloud according to the following instruction: "What is this number?"

**Array-to-Number Word Task:** The children viewed an array (1.5 cm in diameter and 0.5 cm apart) per trial in a rectangle at the bottom of the paper. They were to say the number of dots when the experimenter pointed to the array and asked, "How many dots are here?"

**Number Word-to-Digit Task.** The children heard one number word per trial for each instruction type. They were required to select the matching digit from the lower rectangle that contained six alternative digits from 1 to 6 (no.60 font), according to the instructions: "See which is *x* (a number word 1–6), please point it out with your finger." Number words were verbally repeated twice to lessen the load on children's working memory (this was the same as in the number word-to-array task) [8].

**Number Word-to-Array Task.** The children heard one number word per trial for each instruction type. They were required to select the matching array from six alternative arrays consisting of 1–6 dots (0.6 cm in diameter and 0.1 cm apart) in a small black box (2.6 x 2.6 cm), according to the instructions: "See which box contains only *x* (a number word 1–6) dot(s). Please point it out with your finger."

**Digit-to-Array Task:** The children saw one digit per trial (no.150 font) in a rectangle at the top of the paper. They were required to select the corresponding array from six alternative arrays consisting of 1–6 dots (same as in the number word-to-array task), according to the instructions: "Look at the number above and choose which box below contains the same number of dots. Please point it out with your finger."

**Array-to-Digit Task:** The children saw one array (1.5 cm in diameter and 0.5 cm apart) per trial in a rectangle at the top of the paper. They were required to select the corresponding digit in the lower rectangle containing six alternative

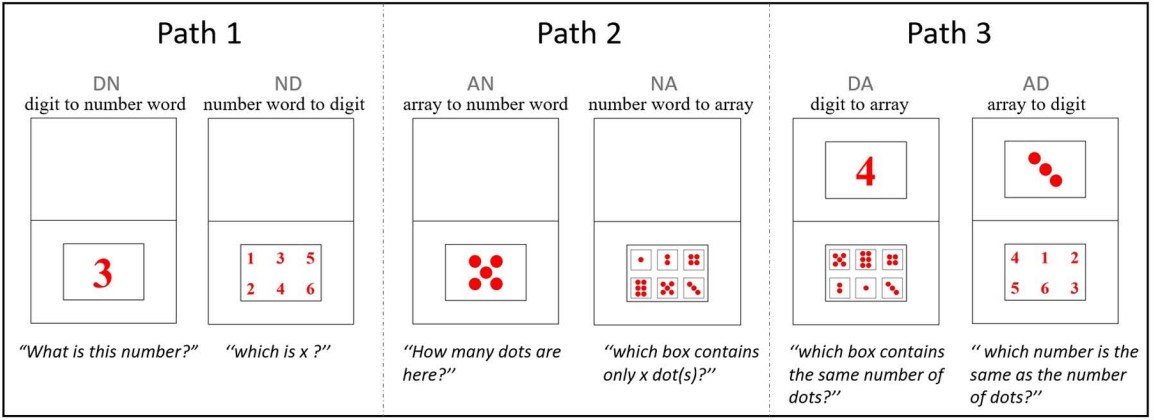

**Fig 1.  Example of visual display of the six mapping tasks.**

digits from 1 to 6 (no. 60 font) according to the instructions: "Look at the dots above and choose which number below is the same as the number of dots. Please point it out with your finger."

**Procedure.** Children were tested individually in a quiet area of their schools. The procedure included three phases: familiarization, practice, and testing. During the familiarization phase, the experimenter first asked the children their names and ages and then asked if they could play a game. With the children's consent, the experimenter and children began to play a game to identify five types of animals: a bird, an elephant, a panda, a squirrel, and a hedgehog. The purpose of the familiarization phase was to make the experimenter and children familiar with each other and build the children's enthusiasm. In the practice phase, the children performed six mapping tasks for numbers 1 and 3, with one trial for each number under each mapping condition, and 12 trials in total. Feedback on the response accuracy was provided only for each practice trial. The practice phase aimed to allow the children to learn how to operate the tasks. To grasp the entire picture of the numerical mapping development of children, every child entered the test phase, even if some scored zero in the practice phase. In the testing phase, each number from 1 to 6 has 2 trials in each task, totaling 72 trials. For each trial, a correct response was scored "1", and an incorrect response or no response was scored "0". Each child participated in all six tasks in the order shown in the tasks part. Each task ended when the child failed in all trials with small numbers. The entire procedure was completed within 10–15 min.

To control for potential confounding variables and enhance the internal validity of the experiment, several measures were implemented. All visual stimuli—including digits and dot arrays—were standardized in font type, size, color, and spatial layout across trials to eliminate perceptual biases. The sequence of target numbers was fixed based on the first two lines of a Latin square design to reduce order effects. In the tasks involving six alternative response options, the position of the alternatives was randomly assigned to prevent positional bias. Additionally, a standardized procedure was followed, with all participants receiving identical verbal instructions administered by a trained experimenter. A structured practice phase with feedback was also included to ensure participants understood the task requirements before proceeding to the formal testing phase.

## Results and discussion

### Difference test of mapping direction

Owing to the non-normal distribution of the children's scores for the numerical mapping tasks, we conducted nonparametric tests. The Wilcoxon signed-rank test was used to explore the differences in the mapping direction of the three representation pairs. We referred to the analytical method of Benoit et al. (2013) [5], and the results are shown in Tables 1–3. These tests showed no reliable differences between the mapping directions of three mapping paths [5]. Although 2-year-old children exhibited significant differences in the mapping direction between digits and arrays for small numbers, this significant difference disappeared when number size and age collapsed. Therefore, to maintain consistency with Benoit et al.'s (2013) analysis and to facilitate comparability across studies, we also collapsed across mapping directions and analyzed only the three primary mapping paths in subsequent analyses.

However, Jiménez Lira et al. (2017) found significant differences between the mapping directions of paths 2 and 3 [8]. Notably, these two mapping paths both involved arrays. For example, in an AN mapping task, children were required to directly determine the number of a square array, whereas in the NA mapping task, children were only required to choose one of two square arrays according to verbal number words. Choosing one from the other was easier than directly stating the number of squares, which may explain the significant differences between mapping directions.

In this study, children were asked to choose one of six candidate arrays in the NA mapping [5], which increased the cognitive demands of the task (e.g., visual scanning and comparison). This elevated difficulty may have made the NA task comparable in challenge to the AN task, where children needed to directly determine the number represented by a single dot array. This design might explain the lack of a significant difference between NA and AN mapping performance observed in our results. Hurst et al. (2017) discovered that children exhibited slightly better AN than NA mapping



**Table 1. Wilcoxon test of the mapping directions for number size.**

| Age | Small numbers (1–3) | | | | Large numbers (4–6) | | | |
|---|---|---|---|---|---|---|---|---|
| | M1 | M2 | Z | p | M1 | M2 | Z | p |
| DN-ND | | | | | | | | |
| 2 | 1.95 | 1.85 | −0.536 | 0.592 | 1.20 | 1.35 | −1.857 | 0.063 |
| 3 | 3.69 | 3.64 | −0.084 | 0.933 | 2.31 | 2.60 | −1.487 | 0.137 |
| 4 | 5.78 | 5.78 | 0.000 | 1.000 | 5.30 | 5.41 | −1.414 | 0.157 |
| AN-NA | | | | | | | | |
| 2 | 1.68 | 1.55 | −0.424 | 0.672 | 0.05 | 0.08 | −1.000 | 0.317 |
| 3 | 4.78 | 4.64 | −0.714 | 0.475 | 1.60 | 1.42 | −0.473 | 0.636 |
| 4 | 5.92 | 5.86 | −0.816 | 0.414 | 4.51 | 4.00 | −1.690 | 0.091 |
| DA-AD | | | | | | | | |
| 2 | 0.65 | 0.33 | −2.020* | 0.043 | 0.00 | 0.03 | −1.000 | 0.317 |
| 3 | 3.13 | 2.82 | −1.509 | 0.131 | 1.31 | 1.38 | −0.082 | 0.934 |
| 4 | 5.46 | 5.65 | −1.823 | 0.068 | 3.62 | 3.76 | −0.528 | 0.597 |

(*** p<0.001; ** p<0.01; * p<0.05, the same below)

*Note:* Small numbers (1–3), large numbers (4–6). M1 refers to the first direction of each mapping path (e.g., DN, AN, DA), and M2 refers to the second direction (e.g., ND, NA, AD). Mapping path abbreviations are as follows: DN = digit-to-number word, ND = number word-to-digit, AN = array-to-number word, NA = number word-to-array, DA = digit-to-array, and AD = array-to-digit. The same definitions and notation apply to Tables 2 and 3.

**Table 2. Wilcoxon test of the mapping directions after collapsing number size.**

| Age | M1 | M2 | Z | p |
|---|---|---|---|---|
| DN-ND | | | | |
| 2 | 3.15 | 3.20 | −0.052 | 0.958 |
| 3 | 6.00 | 6.24 | −0.920 | 0.357 |
| 4 | 11.08 | 11.19 | −1.300 | 0.194 |
| AN-NA | | | | |
| 2 | 1.73 | 1.62 | −0.302 | 0.763 |
| 3 | 6.38 | 6.07 | −0.894 | 0.371 |
| 4 | 10.43 | 9.86 | −1.955 | 0.051 |
| DA-AD | | | | |
| 2 | 0.65 | 0.35 | −1.880 | 0.060 |
| 3 | 4.44 | 4.20 | −1.240 | 0.215 |
| 4 | 9.08 | 9.41 | −1.543 | 0.123 |

**Table 3. Wilcoxon test of the mapping directions after collapsing number size and age.**

| | M1 | M2 | Z | p |
|---|---|---|---|---|
| DN-ND | 6.61 | 6.75 | −1.172 | 0.241 |
| AN-NA | 6.08 | 5.76 | −1.820 | 0.069 |
| DA-AD | 4.61 | 4.52 | −0.943 | 0.346 |

performance [7]. They attributed this to the children being overwhelmed and less likely to count five dot arrays in the NA mapping task (only an average of 9% of trials with counting) compared to the AN mapping task involving only a single dot array (average of 56% of trials with counting). Similarly, as shown in Table 3, children performed slightly better on AN mapping than on NA mapping ($Z = 1.820$, $p = 0.069$). This is one reason why we introduced a condition to remind children to count in Experiment 2.

### Comparison of mapping scores for different ages

To explore whether children's mapping abilities for each mapping path developed significantly from 2 to 4 years of age. Each mapping path was analyzed separately using nonparametric tests. The Kruskal-Wallis test indicated significant differences across the three age groups for each mapping path (path 1: $H = 48.436$, $p < 0.001$; path 2: $H = 78.032$, $p < 0.001$; path 3: $H = 74.221$, $p < 0.001$). Results of the multiple comparisons with Bonferroni correction revealed that the mean numerical mapping scores of 2-year-old children were significantly lower than those of 3-year-olds (path 1: $p < 0.05$; paths 2 and 3: $ps < 0.001$) and 4-year-olds ($ps < 0.001$). Additionally, 3-year-olds scored significantly lower than 4-year-olds across all paths ($ps < 0.001$). These findings indicated a significant age-related improvement in numerical mapping performance across all mapping paths [5,7–9].

### Comparison of mapping scores for small and large numbers

To explore the differences in children's mapping performance for number size, we analyzed each mapping path separately using the Wilcoxon signed-rank test to compare the mapping scores for small and large numbers. The findings indicated that children's mapping performance was better for small than for large numbers across all three mapping paths (path 1: $Z = -5.930$, $p < 0.001$; path 2: $Z = -7.790$, $p < 0.001$; path 3: $Z = -6.628$, $p < 0.001$). Unsurprisingly, this result is consistent with those of previous studies [5,7–9].

### Test of mapping performance above chance

To determine if children's mapping performance significantly exceeded the chance level, we employed the Wilcoxon signed-rank test to compare mapping scores with an expected value of one out of six (representing the six potential response options per trial) (Table 4).

For 4-year-old children, their performance was consistently above chance regardless of the number size and mapping pairs [5,7]. For 3-year-old children, the performance on mapping paths 2 and 3 for large numbers was below the chance

**Table 4. Wilcoxon test of mapping performance above chance.**

| Age | Mapping pairs | Small numbers | | Large numbers | |
|---|---|---|---|---|---|
| | | Z | p | Z | p |
| 2 | path 1 | 1.328 | 0.184 | −1.214 | 0.225 |
| | path 2 | 1.193 | 0.233 | −5.932 | < 0.001 |
| | path 3 | −3.654 | < 0.001 | −6.252 | < 0.001 |
| 3 | path 1 | 5.140*** | < 0.001 | 2.783** | 0.005 |
| | path 2 | 5.909*** | < 0.001 | 1.347 | 0.178 |
| | path 3 | 4.283*** | < 0.001 | −0.215 | 0.830 |
| 4 | path 1 | 5.878*** | < 0.001 | 5.582*** | < 0.001 |
| | path 2 | 5.822*** | < 0.001 | 4.985*** | < 0.001 |
| | path 3 | 5.639*** | < 0.001 | 4.388*** | < 0.001 |

Note: path 1 (digit/number word), path 2 (array/number word), and path 3 (digit/array); Chance level = 1.

level. However, Benoit et al. (2013) found that the performance on paths 1 and 3 for small numbers, as well as performance on paths 1, 2, and 3 for large numbers, was all below the chance level [5]. These results differ from those reported by Benoit et al. (2013), in which 3-year-old children's performance on most mapping paths was below chance. While both studies used similar materials and procedures, the performance differences may reflect variability in children's numerical development across samples or differences in educational or cultural context.

In addition, Hurst et al. (2017) conducted research with American children aged 3-to-4 years and found that only 3-year-old children's performance on path 3 for large numbers (4–5) was below the chance level [7]. This difference may be because our study added the number "6" to the large number range, thereby increasing the difficulty of numerical mapping. Additionally, they provided five options for verbal number words in AN mapping, which is relatively simple compared to directly determining the number of arrays [7]. In contrast, in this study, the performance of 3-year-old children on path 2 with large numbers remained below the chance level due to the expansion of the large number range and the absence of options.

For 2-year-old children, their mapping performance was not significantly above chance in any conditions (either below or not significantly above the chance level). The main reason for this was that 19 out of 40 2-year-old children (47.5%) scored zero on all mapping tasks. Therefore, to master the mapping performance of 2-year-old children who have started to develop mapping abilities, we adopted a two-step approach in the analysis. First, we included all participants, including those who scored zero on all tasks (see Table 4). Then, we conducted a supplementary analysis excluding children with zero scores, as this allowed us to further explore the performance of children who demonstrated mapping abilities (see Table 5). This second analysis was performed using the Wilcoxon test. The results showed that children's mapping performance was above the chance level on paths 1 and 2 for small numbers.

Overall, the 4-year-olds had developed all six mapping abilities, including small and large numbers. The 3-year-olds had developed all six mapping abilities for small numbers and had started to understand how to map in path 1 for large numbers. Notably, approximately half of the 2-year-olds learned to map in paths 1 and 2 for small numbers.

## Developmental order of the three mapping pairs

To examine performance differences between the three numerical mapping paths within each age group, we conducted pairwise comparisons using the Wilcoxon signed-rank test. Bonferroni correction was applied by multiplying the original $p$-values by the number of comparisons (i.e., ×3), and the adjusted $p$-values are reported below.

As shown in the Age 2 section of Fig 2, 2-year-old children began to develop the ability to map in paths 1 and 2 for small numbers, as demonstrated by the mapping test performance above chance. However, in this experiment, only 21 out of 40 2-year-old children (52.5%) had begun to acquire this ability. Furthermore, the Wilcoxon test results showed no significant differences in performance between paths 1 and 2 for small numbers ($Z = 0.809$, $p = 1.257$).

The results for 3-year-olds are presented in the middle of Fig 2. For small numbers, the Wilcoxon test revealed significantly better performance on path 2 compared to both path 1 ($Z = 3.676$, $p < 0.001$) and path 3 ($Z = 4.854$, $p < 0.001$), and better performance on path 1 than on path 3 ($Z = 3.623$, $p < 0.001$). This indicated that, for 3-year-old children, the

**Table 5. Wilcoxon test of 2-year-old children's mapping performance above chance after excluding children with zero scores.**

| Mapping pairs | Small numbers | | | Large numbers | |
|---|---|---|---|---|---|
| | $Z$ | $p$ | | $Z$ | $p$ |
| path 1 | 3.616*** | < 0.001 | | 1.326 | 0.185 |
| path 2 | 3.444** | 0.001 | | −4.044 | < 0.001 |
| path 3 | −1.153 | 0.249 | | −4.491 | < 0.001 |

Note: path 1 (digit/number word), path 2 (array/number word), and path 3 (digit/array); Chance level = 1

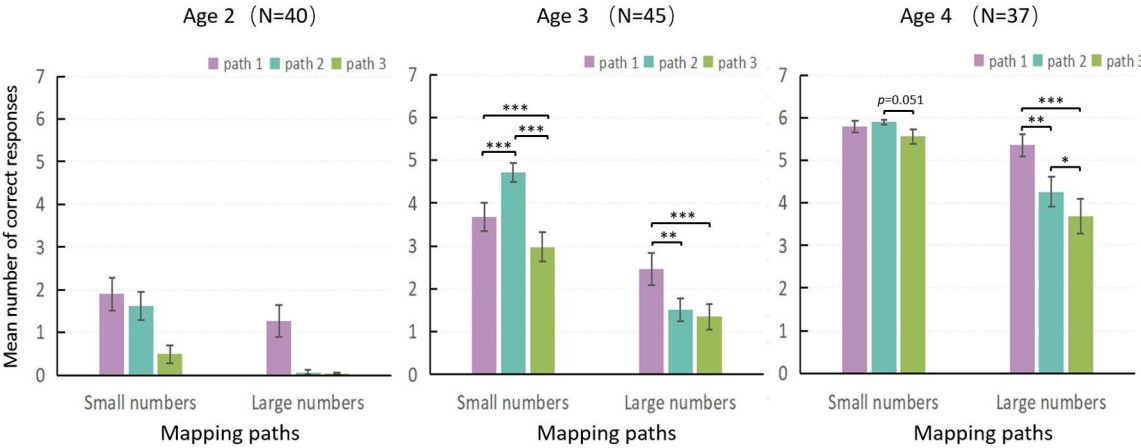

**Fig 2. Mean number of correct responses for paths 1, 2, and 3, by number size.** *Note: path 1 (digit/number word), path 2 (array/number word), and path 3 (digit/array).*

developmental order for small numbers was: 2→1→3. For large numbers, the performance on path 1 was significantly better than that on either path 2 ($Z = 2.999$, $p < 0.01$) or path 3 ($Z = 3.829$, $p < 0.001$). However, no significant differences were found between the latter two mapping pairs ($Z = 0.894$, $p = 1.113$). Therefore, the development of path 1 preceded that of paths 2 and 3 for large numbers (1→2 and 3).

The results for 4-year-olds are presented on the right side of Fig 2. For small numbers, no significant differences were found between path 1 and path 2 ($Z = 0.707$, $p = 1.440$) or between path 1 and path 3 ($Z = 1.973$, $p = 0.147$). Only the comparison between path 2 and path 3 showed a marginal difference, with slightly better performance on path 2 ($Z = 2.388$, $p = 0.051$). These results suggest that by age 4, children's performance on small-number mappings had approached ceiling levels, resulting in minimal differences among the three paths. For large numbers, the Wilcoxon test revealed significantly better performance on path 1 compared to both path 2 ($Z = 2.962$, $p < 0.01$) and path 3 ($Z = 3.776$, $p < 0.001$), and better performance on path 2 than on path 3 ($Z = 2.745$, $p < 0.05$). This indicated that, for 4-year-old children, the developmental order for large numbers was 1→2→3.

In conclusion, developmental orders differed between small and large number mapping. The results indicated that learning mapping path 3 was the most difficult for children [7,8,18,20]. In daily life, children usually encounter connections between number words and digits without considering the non-symbolic numbers they represent. For instance, children often see "3" and then say "three", even if no actual objects correspond to the number [7]. Therefore, it could be challenging for children to learn how to connect digits with arrays (path 3). In addition, the difference in developmental order between small and large numbers is the positional relationship between paths 1 and 2 (See General discussion for cause analysis).

In Experiment 2, we introduced a condition to remind participants to count to explore how this influenced numerical mapping. Two-year-old children were not considered in Experiment 2 because most of them could not yet perform simple counting.

## Experiment 2

### Method

**Participants.** To ensure that the socioeconomic status of participants in Experiment 2 was comparable to that in Experiment 1, we recruited 97 children aged 3–4 years from three preschools located in Lugu Town, Liangshan Prefecture, Sichuan Province. All participating preschools were inclusive, non-profit institutions, with standardized tuition



fees set by local government regulations, ranging from CNY 1,600–2,000 per semester. These schools served children from local families and did not apply any selective admission policies. Informal information obtained at the school level further suggested that most parents shared similar occupational backgrounds, primarily in small business, service industries, or skilled labor. After excluding four children (1 three-year-old and 3 four-year-olds) who did not undergo the entire procedure, 93 participants remained. We divided the participants into two age groups: 49 three-year-olds (20 male; range = 36–47 months; M = 3 years 6 months) and 44 four-year-olds (21 male; range = 48–58 months; M = 4 years 4 months).

The experiment was conducted from December 11th, 2023, to January 4th, 2024, while Experiment 1 was carried out from April to May 2023. Although the two experiments were conducted in different months, both involved children of similar ages and focused on foundational numerical cognition [2], which is generally stable and less influenced by seasonal or environmental changes. Given this, we believe the time gap between the two experiments is unlikely to have introduced any significant bias.

**Procedure.** Based on Experiment 1, Experiment 2 introduced a condition to remind children to count. In the AN, NA, AD, and DA tasks, the experimenter gave a standardized instruction before the child responded, saying, "Please count using your hand. How many dots are there?" To ensure consistency across participants, the experimenter instructed each child to use their right index finger to point at the dots one by one while verbally counting them aloud. This procedure was strictly followed to standardize the counting reminder across all children and trials.

## Results and discussion

### Difference test of mapping direction

To investigate whether reliable differences could be found between the mapping directions of the three representation pairs, we conducted a Wilcoxon signed-rank test. For 3-year-old children, the result showed significant differences in the mapping direction of paths 2 and 3 for larger numbers. For 4-year-old children, marginally significant differences were found between these two mapping pairs (Table 6). One possible explanation for these differences is the use of counting [7] (see General discussion).

### Test of mapping performance above chance

To determine whether children's mapping performance was reliably above the chance level, we used the Wilcoxon signed-rank test. Only 3-year-old children's performance on path 3 with large numbers was at chance level ($Z = 1.288$, $p = 0.198$),

**Table 6. Wilcoxon test of the mapping directions by number size.**

|  | Small numbers | | | | | Large numbers | | | |
| --- | --- | --- | --- | --- | --- | --- | --- | --- | --- |
| Age | M1 | M2 | Z | p | | M1 | M2 | Z | p |
| DN-ND |  |  |  |  |  |  |  |  |  |
| 3 | 4.06 | 4.29 | −1.081 | 0.280 | | 3.04 | 3.22 | −1.238 | 0.216 |
| 4 | 5.55 | 5.55 | 0.000 | 1.000 | | 4.93 | 5.11 | −1.191 | 0.234 |
| AN-NA |  |  |  |  |  |  |  |  |  |
| 3 | 5.41 | 5.31 | −0.314 | 0.753 | | 3.80 | 2.57 | −3.178* | 0.001 |
| 4 | 5.95 | 5.86 | −1.414 | 0.157 | | 5.68 | 5.34 | −1.911 | 0.056 |
| DA-AD |  |  |  |  |  |  |  |  |  |
| 3 | 3.90 | 3.73 | −1.001 | 0.317 | | 1.43 | 2.43 | −3.381* | 0.001 |
| 4 | 5.36 | 5.50 | −1.098 | 0.272 | | 4.61 | 4.93 | −1.922 | 0.055 |

*Note: Small numbers (1–3), large numbers (4–6). M1 refers to the first direction of each mapping path (e.g., DN, AN, DA), and M2 refers to the second direction (e.g., ND, NA, AD). Mapping path abbreviations are as follows: DN = digit-to-number word, ND = number word-to-digit, AN = array-to-number word, NA = number word-to-array, DA = digit-to-array, and AD = array-to-digit.*

which was consistent with the results of Hurst et al. (2017) [7]. However, in Experiment 1, the mapping performance of 3-year-old children for both paths 2 and 3 for large numbers was at chance level. This indicated that reminding children to count could significantly improve their mapping performance on path 2.

## Comparative analysis under two conditions

Based on the data from Experiments 1 and 2, 3-to-4-year-old children's mapping performance was compared on paths 2 and 3 under the two conditions: not reminding to count and reminding to count (Fig 3).

For 3-year-old children, the results of the Mann–Whitney U test indicated that performance on path 2 was significantly better with reminders than without reminders, regardless of whether the numbers were small ($Z=−2.454$, $p<0.05$) or large ($Z=−3.373$, $p<0.01$). In addition, performance on path 3 for small numbers was slightly better with reminders than without reminders ($Z=−1.793$, $p=0.073$), suggesting a trend toward improvement despite the high cognitive demand of this task at age 3. For 4-year-old children, performance on paths 2 and 3 for large numbers was significantly better when they were reminded to count than when they were not (path 2: $Z=−3.070$, $p<0.01$; path 3: $Z=−2.234$, $p<0.05$). However, for small numbers, performance on both paths 2 and 3 was already close to ceiling levels, which may have limited the observable effects of reminders in these conditions. Therefore, it can be concluded that in numerical mapping tasks, reminding children to count can improve their mapping performance on paths 2 and 3, especially when the tasks are relatively difficult and children's baseline performance leaves more room for improvement.

## Developmental order of three mapping pairs

As in Experiment 1, pairwise comparisons were performed using the Wilcoxon signed-rank test with Bonferroni correction applied for multiple comparisons.

The results for the 3-year-olds are presented on the left side of Fig 4. For small numbers, performance on path 2 was significantly better than on path 1 ($Z=4.180$, $p<0.001$) and path 3 ($Z=4.714$, $p<0.001$), while no significant difference was found between paths 1 and 3 ($Z=2.071$, $p=0.114$). Therefore, it could be inferred that, for 3-year-old children, the

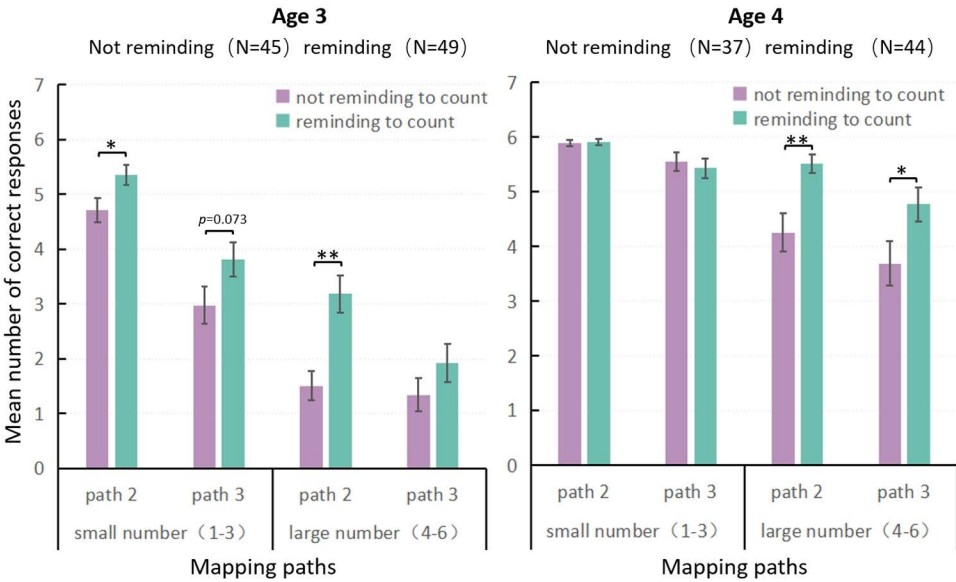

**Fig 3. The differences in numerical mapping scores under (not) reminding to count.** *Note: path 1 (digit/number word), path 2 (array/number word), and path 3 (digit/array).*

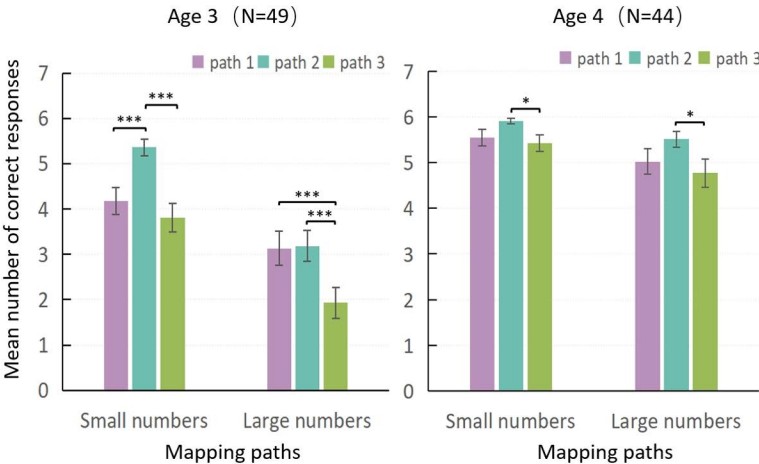

**Fig 4. Mean number of correct responses for mapping paths 1, 2, and 3 with reminding children to count, by number size.** *Note: path 1 (digit/ number word), path 2 (array/number word), and path 3 (digit/array).*

developmental order for small numbers was 2→1 and 3. Compared with the result (2→1→3) in Experiment 1, the mapping performance on path 3 caught up with that on path 1, indicating that reminding children to count promoted mapping performance on path 3. For large numbers, children's performance on path 1 was better than that on path 3 ($Z=4.430$, $p<0.001$), and that on path 2 was better than that on path 3 ($Z=4.164$, $p<0.001$). However, no significant differences were found in performance between paths 1 and 2 ($Z=0.165$, $p=2.607$) (2 and 1→3). Compared with the result (1→2 and 3) in Experiment 1, the mapping performance on path 2 caught up with that on path 1, indicating that reminding children to count promoted mapping performance on path 2.

The results for 4-year-olds are shown on the right side of Fig 4. For small numbers, the Wilcoxon test revealed significantly better performance on path 2 than on path 3 ($Z=2.587$, $p<0.05$). However, no significant differences were found between path 1 and path 2 ($Z=2.100$, $p=0.108$) or between path 1 and path 3 ($Z=0.742$, $p=1.374$). Therefore, although a partial trend favoring path 2 over path 3 was observed, no clear developmental order could be established among the three mapping paths—a pattern consistent with the findings of Experiment 1. For large numbers, the Wilcoxon test revealed significantly better performance on path 2 than on path 3 ($Z=2.975$, $p<0.01$). However, no significant differences were found between path 1 and path 2 ($Z=1.762$, $p=0.234$), or between path 1 and path 3 ($Z=1.264$, $p=0.618$). Compared to the result of Experiment 1 (1→2→3), the current findings suggest that performance on path 2 has caught up with path 1, while performance on path 3 has also improved, approaching the level of path 1. These changes indicated that reminding children to count may have facilitated improved mapping performance on both path 2 and path 3.

As shown in Table 7, under the condition of not reminding to count, developmental orders differed between small and large number mapping regardless of age. Under the condition of reminding to count, 3-year-olds still exhibited different developmental orders for small and large numbers. Although, by the age of 4, the developmental orders for small and large numbers began to converge. On the whole, the developmental orders for small and large numbers remained distinct.

## General discussion

The current study mainly aimed to explore the developmental characteristics of children's numerical mapping for small and large numbers under two conditions: not reminding them to count and reminding them to count. The results indicated that the developmental orders of the numerical mapping paths differed for small and large numbers. Meanwhile, reminding



**Table 7. Comparison of the developmental orders under two conditions by number size.**

| Condition | Age | small numbers | large numbers |
|---|---|---|---|
| not reminding to count | 3 | 2→1→3 | 1→2 and 3 |
| | 4 | No clear order (partial: 2>3) | 1→2→3 |
| reminding to count | 3 | 2→1 and 3 | 2 and 1→3 |
| | 4 | No clear order (partial: 2>3) | No clear order (partial: 2>3) |

*Note: path 1 (digit/number word), path 2 (array/number word), and path 3 (digit/array). "No clear order (partial: 2 > 3)": Although a partial trend favoring path 2 over path 3 was observed, no significant differences were found between path 1 and path 2, or between path 1 and path 3. Therefore, no clear developmental order could be established among the three mapping paths.*

children to count improved their mapping performance, leading to changes in the symmetry of mapping direction and developmental order of mapping paths.

## Numerical mapping direction

In Experiment 1, without reminding the children to count, no significant differences were found in the mapping direction among the three mapping paths, which is consistent with the results of Benoit et al. (2013) [5]. However, in Experiment 2, when children were reminded to count, 3-year-olds showed significant differences in the mapping direction on paths 2 and 3 for large numbers, while 4-year-olds showed marginally significant differences in the mapping direction on paths 2 [7] and 3 for large numbers. Furthermore, the results showed that the AN and AD mappings had higher scores than their inverse mappings (NA and DA). These findings indicated that reminding children to count can change the symmetry of the mapping direction for paths 2 and 3.

Researchers have suggested that counting is not elicited equally across distinct contexts [64,65], leading to differences in mapping performance [7]. When they were not reminded to count in the AN and AD mapping tasks for large numbers, some children who had learned to count might obtain the answer by estimating target arrays instead of actively counting. Thus, the scores were similar to those for the inverse mappings, and with no significant differences in mapping directions. However, when reminding them to count, children could obtain the answer by counting only an array in the AN and AD mapping tasks for large numbers. In contrast, in the NA and DA mapping tasks, the children needed to choose an array from six candidate arrays. Counting six candidate arrays simultaneously can be time-consuming and laborious [7], resulting in relatively low scores. Thus, significant differences were found in the mapping directions on paths 2 and 3 for large numbers.

## Numerical mapping level

In Experiment 1, for children aged 3-to-4 years, the average scores on path 1 were higher than those of French children in the previous study [5]. Specifically, 3-year-old children in this study (small numbers: M = 3.67, large numbers: M = 2.46) were found to be at a developmental level similar to that of 4-year-old children in that study (small numbers: M = 4, large numbers: M = 2.13) [5]. Furthermore, 4-year-old children in this study (small numbers: M = 5.78, large numbers: M = 5.36) showed almost equivalent performance to 5-year-old children in that study [5] (performing perfectly on all tasks as described) on path 1. These differences may be attributed to variations in socioeconomic status or differences in educational backgrounds [10].

Previous studies have mainly explored the development of numerical mapping from the age of 3 [5,7]. Although some studies recruited 2-year-old participants, they did not discuss the developmental status of the 2-year-old age group

separately from the others [8,9]. Therefore, we specifically recruited 2-year-old children as a group to explore their developmental status in the current study. The results showed that around half of the 2-year-olds had already started learning digit-number word and array-number word mapping for small numbers. This indicated that the 2-year-old children are in the embryonic stage of numerical mapping.

Subsequently, we comprehensively considered the results of both experiments and found that reminding children to count could significantly improve their performance in mapping paths 2 and 3. Scores of 3-year-olds on path 2 for large numbers were found to be significantly higher than chance level when they were reminded to count, but not significantly higher than chance level when reminders were not provided. Furthermore, 3-year-old children's scores on path 2 were significantly higher with reminders than without, and children's scores on path 3 with reminders for small numbers were also marginally higher than those without reminders. In addition, 4-year-olds' scores on paths 2 and 3 for large numbers were significantly higher with reminders than without.

Therefore, reminding children to count could significantly improve their performance on paths 2 and 3 [51,53]. This supported in reverse the viewpoint that, without reminding children to count, array-related numerical tasks will underestimate their number concept level [25,24, 62,63].

## Developmental order of the three mapping paths

Although previous studies also classified the size of numbers, they did not explore the developmental orders of numerical mapping with small and large numbers separately [5,7,9]. Therefore, they did not find differences in the developmental orders of numerical mapping for small and large numbers. However, in this study, we found they were different (see Table 7).

First, it can be inferred that learning path 3 mapping is most difficult for children, regardless of the number size [7–9,18,20]. Bialystok (1992) suggested that children acquire the ability to map in path 3 last because they first need to understand the concept that each number word corresponds only to a specific digit and a unique array [18]. That is, understanding the concept of cardinality is a prerequisite for mapping path 3 [20,69]. The intermediate analysis revealed that children initially cannot directly map in path 3, and instead use path 1 as the intermediary [7]. Specifically, once children form the ability to map between symbolic numbers (number words) and non-symbolic numbers (arrays), they can utilize this learned mapping to comprehend path 3 by mapping the second symbolic numbers (digits) back to the existing symbolic numbers (number words). The "symbolic account" also indicated that path 1 serves as an intermediary [7,9,70], helping to explain why children find it challenging to map in path 3.

However, this observed pattern contrasts sharply with the findings of Benoit et al. (2013) [5], who demonstrated that French children were able to map in path 3 before mastering path 1. They attributed the reasons to the privileged status of non-symbolic code and the permanently available presentations of arrays and digits. However, similar results did not appear in this study, despite using almost identical materials. Notably, both studies presented non-symbolic numbers in a familiar format (pattern on dice). The symbolization of the dot array arrangements may provide an advantage for numerical mapping in French children, leading to better performance on path 3 mapping [5,23]. However, the children in this study performed poorly in mapping path 3, indicating that most of them were unfamiliar with the dot arrangement mode or had not yet symbolized it.

Second, the difference in developmental order between small and large numbers is the positional relationship between paths 1 and 2. For a small number range, the mapping performance was better on path 2 than on path 1 under two conditions. For a large number range, the mapping performance was better on path 1 than on path 2 without counting reminders, but the situation reversed with counting reminders. This may confirm two mechanisms of number representation: the OTS for small numbers and the ANS for large numbers [36,41,42,46]. Small numbers belong to the subitizing number range (1–3 or 4), in which children can obtain the accurate number of an array quickly through "subitizing" [47]. Therefore, children find it easier to learn path 2 for small numbers, leading to the mapping performance was better on path 2 than on path 1.

However, when learning to map in path 2 with large numbers (greater than 3), children often have difficulties in obtaining an accurate number of an array by ANS or in spontaneously using counting strategies to map accurately in path 2 without counting reminders [7,64,65]. In contrast, around age 3, children in this study begin to learn Arabic digits within 10. Therefore, in the range of large numbers, the mapping performance on path 1 was found to exceed that on path 2. But then, when the children were reminded to count, the performance on path 2 surpassed or caught up with path 1. These results provided a new insight that both the number size and whether to remind children to count can affect the judgment on the developmental order of numerical mapping.

Over all, based on previous debates, we examined the symmetry of the mapping direction and developmental order of mapping paths from the perspectives of number size and whether to remind children to count and found that these two factors would affect the assessment of the development order of numerical mapping. Those provided two pieces of empirical evidence that explain the differences found in earlier studies, as well as a reference for the cultivation and assessment of preschool children's numerical mapping ability. In particular, it was observed that the developmental orders of small- and large-number mapping differed, supporting two potential mechanisms: an object tracking system and an approximate number system. Furthermore, it was found that the enlightenment period of children's numerical mapping development occurs around the age of 2, also offering insights for cultivation of preschool children's numerical mapping ability.

## Conclusion

First, approximately half of the 2-year-old children had already started learning digit-number word and array-number word mapping for small numbers. Second, the developmental orders of three numerical mapping paths differed between small and large numbers. Third, reminding children to count could improve their performance on array-number word and array-digit mappings, resulting in alterations in the symmetry of the mapping direction and developmental order of mapping paths. In conclusion, number size and counting reminders play significant roles in shaping researchers' assessments of developmental order of numerical mapping in preschool children.

### Limitations and future directions

This study has several limitations. First, the participants were recruited from three towns across two provinces, which may limit the generalizability of the findings. The study also employed a cross-sectional design with a fixed task order, potentially introducing order effects and individual variability. Future research should consider broader sampling, longitudinal tracking, and fully randomized within-subject designs to enhance generalizability and experimental control.

Second, the use of traditional paper-and-pencil assessments limited the ability to capture reaction times. Incorporating digital touch-screen tasks could enable more precise measurement, particularly for preschool children, and support more interactive testing paradigms.

Third, the asymmetry in forward and backward mapping tasks involving number words may have introduced differences in cognitive demands. Although this design aligned with prior studies (e.g., Benoit et al., 2013), future research should adopt more balanced task structures. Additionally, although the effects of counting reminders were analyzed within each age group, counting strategies may still interact with cognitive resources such as working memory, warranting further investigation.

### Acknowledgments

We appreciate the participants, their parents or guardians, and the school leaders who supported this study.

### Author contributions

**Conceptualization:** Jun Zhu, Huanyu Yang, Fangwen Yu, Yun Pan.

**Data curation:** Jun Zhu.



**Formal analysis:** Jun Zhu, Liangzhi Jia, Qiang Wu.

**Funding acquisition:** Yun Pan.

**Investigation:** Jun Zhu, Chenli Li, Qiang Wu.

**Methodology:** Jun Zhu, Huanyu Yang, Liangzhi Jia, Fangwen Yu, Yun Pan.

**Project administration:** Yun Pan.

**Resources:** Jun Zhu, Chenli Li.

**Software:** Jun Zhu.

**Supervision:** Yun Pan.

**Validation:** Jun Zhu, Huanyu Yang.

**Visualization:** Jun Zhu, Huanyu Yang, Liangzhi Jia, Fangwen Yu.

**Writing – original draft:** Jun Zhu.

**Writing – review & editing:** Jun Zhu, Huanyu Yang, Liangzhi Jia, Chenli Li, Yajie Bi, Fangwen Yu, Yun Pan.

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
