## [Decision Letter · Decision Letter 0]

PONE-D-25-02198The development of numerical mapping in preschool childrenPLOS ONE

Dear Dr. Pan,

Thank you for submitting your manuscript to PLOS ONE. After careful consideration, we feel that it has merit but does not fully meet PLOS ONE’s publication criteria as it currently stands. Therefore, we invite you to submit a revised version of the manuscript that addresses the points raised during the review process.

ACADEMIC EDITOR: 

Although the manuscript is well written, it has several methodological drawbacks that need to be addressed.Please check out all reviewers' comments and remarks. 

We look forward to receiving your revised manuscript.

Kind regards,

Alessandro Bruno, Ph.D.

Academic Editor

PLOS ONE

 [This work was supported by the National Natural Science Foundation of China (grant number 32360203); and the Major project of Key Research base of Humanities and Social Sciences of Ministry of Education (grant number 22JJD10009).]. 

4. In the online submission form, you indicated that [All data that support the findings of this study are available from the corresponding author upon reasonable request.].

Additional Editor Comments:

Dear Authors,

Your paper has several major drawbacks.

You're invited to address all reviewers' remarks and comments in order to enhance the overall paper.

Kindest regards,

A.B.

Reviewers' comments:

Reviewer's Responses to Questions

Comments to the Author

1. Is the manuscript technically sound, and do the data support the conclusions?

Reviewer #1: Partly

Reviewer #2: Partly

Reviewer #3: Yes

Reviewer #4: No

Reviewer #5: Partly

2. Has the statistical analysis been performed appropriately and rigorously? 

Reviewer #1: Yes

Reviewer #2: Yes

Reviewer #3: I Don't Know

Reviewer #4: No

Reviewer #5: Yes

3. Have the authors made all data underlying the findings in their manuscript fully available?

Reviewer #1: No

Reviewer #2: No

Reviewer #3: Yes

Reviewer #4: No

Reviewer #5: Yes

4. Is the manuscript presented in an intelligible fashion and written in standard English?

Reviewer #1: Yes

Reviewer #2: Yes

Reviewer #3: No

Reviewer #4: Yes

Reviewer #5: Yes

5. Review Comments to the Author

Reviewer #1: The manuscript presents an interesting investigation; however, several areas require further clarification to strengthen the methodology, enhance transparency, and improve the overall quality of the study.

1. Data Availability

As per PLOS Data Policy, all data underlying the findings should be made available without restriction, unless there are specific reasons for withholding it. The manuscript states that readers must "reach out to the corresponding author with a reasonable request" to obtain the data. Please provide a clear justification for why the data cannot be shared publicly and why restrictions on sharing exist.

2. Age Grouping Justification

The manuscript presents age groupings of 2-, 3-, and 4-year-olds without providing a rationale for these specific categories. It would be beneficial to explain why these particular age groupings were chosen and how they align with relevant developmental stages to support the robustness of the study design.

3. Sampling Strategy

While the geographic locations of participant recruitment are mentioned, the manuscript lacks a description of the sampling strategy. It remains unclear whether this was a convenience sample or if any selection criteria were applied. Please provide more detail on the recruitment process, including whether participants were randomly selected or chosen based on particular inclusion/exclusion criteria. This additional information will help assess potential biases in the sampling method and improve the transparency of the study design.

4. Control Conditions

In Experiment 1, the manuscript does not specify what control conditions were implemented to ensure the results were not influenced by confounding variables. A clearer description of the control measures would improve the interpretability of the results.

5. Participant Selection Rationale:

While the authors state that they aimed to match the socioeconomic status of participants in Experiment 2 to those in Experiment 1, no details are provided on how this was assessed or confirmed. The different geographical location (Liangshan Prefecture vs. Dangwu Town and Mengyang Town in Experiment 1) raises questions about potential regional differences.

6. Time Gap Between Experiments:

Experiment 1 was conducted from April to May 2023, while Experiment 2 was conducted from December 2023 to January 2024. The potential impact of this 7–8-month gap is not addressed in the methodology. Please consider discussing whether this gap might have introduced seasonal or contextual changes that could affect participants’ behaviors or experiences. If the gap is not considered impactful, a justification for this assumption would strengthen the methodology.

7. Sample Representation and Generalizability:

The participants were recruited from five preschools in two cities in China. While this is a reasonable geographical spread, it would be useful to comment on the demographic characteristics of the children. This would help assess the generalizability of the findings beyond this specific sample. The homogeneity of the sample should also be discussed to evaluate the external validity of the study's conclusions.

8. Reminding Children to Count:

The inclusion of the "reminding children to count" condition in Experiment 2 is an interesting and relevant manipulation. However, more details on how the counting reminder was delivered separately and its duration would be helpful for evaluating the effectiveness of this intervention. Was there a standardized method for reminding the children to count, and how was it ensured that the reminder was consistent across participants?

In sum, the manuscript presents an intriguing investigation, but addressing the concerns outlined above—particularly in the areas of data availability, control conditions, participant selection, and recruitment strategy—will significantly improve the clarity and validity of the study. Given these concerns, a major revision is warranted for publication.

Reviewer #2: 1.Introduction Section. The current introduction lacks sufficient elaboration on real-world problems and research background.

2.Literature Review. It is recommended to add a dedicated "Literature Review" section. Reorganize existing content from the Introduction to systematically summarize prior studies with logical taxonomies.

3.Reference Timeliness. Add some relevant studies within five years.

4.Comparative Analysis of Experiments. The comparative analysis between the two experiments is insufficient.

5.Innovation Clarification. The articulation of research novelty remains vague.

Reviewer #3: Benoit et al. (2013) focused on the directionality of children’s numerical mapping (e.g., array-to-digit, digit-to-array, number word-to-digit, digit-to-number word), which suggests that the number of mapping paths is not limited to three (digit-number word, array-number word, digit-array) but rather includes six different mapping directions. In the present study, the classification of mapping paths into three categories needs further justification. If Benoit et al.’s (2013) study supports a six-path distinction, the authors should clarify why they have simplified it to three, or whether they have overlooked potential differences between the various mapping directions.

Additionally, the study suggests that children follow a developmental sequence in numerical mapping (e.g., 2→1→3 or 1→2→3), but it is unclear whether such a sequence is universally applicable. The current research needs to provide stronger theoretical evidence demonstrating that numerical mapping must follow a specific developmental order. Could there be individual differences, or might task types influence the order? If this assumption lacks a solid foundation, it could undermine the study’s framework. Therefore, the authors are advised to clarify the following points:

1. Why classify the mapping paths into three rather than six?

2. What theoretical and empirical evidence supports the developmental order of numerical mapping? Are there studies indicating that children follow a fixed sequence in acquiring numerical mapping skills, or could this sequence be influenced by factors such as culture, educational background, or task type?

3. The relationship between mapping direction and developmental order: If different mapping directions exist, does the developmental path remain consistent? For instance, do array-to-digit and digit-to-array mappings develop simultaneously, or is there a sequential progression?

If sufficient theoretical or empirical support cannot be provided, it is recommended to reconsider the rigid developmental sequence framework to avoid weakening the study’s explanatory power and theoretical contribution.

Reviewer #4: Overall Comment

The manuscript is well-written and generally easy to follow. However, certain methodological details require further clarification, particularly regarding task design and statistical procedures. Since these aspects are fundamental to interpreting the results, providing more detail would strengthen the study’s validity. Consequently, the current conclusions may not be fully supported without further methodological and analysis clarification.

Introduction

Major Concerns

1. Review of Previous Research for Replication (Lines 28–37, 115–119)

The introduction briefly mentions previous studies with differing findings to justify replication. However, a detailed comparison of these studies is necessary to identify potential sources of discrepancies. Otherwise, differences in findings may simply reflect variations in experimental design rather than a meaningful replication attempt.

• The summary suggests that differences in experimental materials (e.g., number size), procedures, or participant demographics (e.g., socioeconomic background, education levels) may contribute to variations in findings. However, these factors are not sufficiently reviewed.

• How do these previous studies compare in terms of similarities and differences? What aspects might have led to different outcomes?

• Based on this review, what specific experimental design choices were made in the current study to ensure replication?

• The manuscript states that it closely follows Benoit et al. (2013) in methodology. If so, why was this particular study chosen? Were there any modifications to the methodology?

2. Review of Previous Research on the Developmental Order of Numerical Mapping (Lines 38–75)

Some explanations in the introduction appear inconsistent with reported findings.

• Benoit et al. (2013) is cited as explaining why children learn Path 1 before Path 3, yet their findings indicate a developmental sequence of 2 → 3 → 1. Could this discrepancy be clarified?

• Hurst et al. (2017) presents two accounts:

o Quantity account: 2 → 3 → 1

o Symbolic account: 2 → 1 → 3

However, their findings indicate 2 & 1 → 3, which contradicts both accounts. Could this be addressed?

• Additionally, why does the manuscript reference Hurst et al. (2017) rather than Benoit et al. (2013) to explain developmental order, despite closely following Benoit et al. (2013) in methodology? Would it be more appropriate to reference Benoit et al. (2013)’s explanations instead? If not, are there limitations in Benoit et al.’s explanation that justified focusing on Hurst et al. (2017)?

• Finally, the second hypothesis is quite general. Could the predicted outcomes be specified in more detail? Given that Path 3 is relatively robust in prior research (except in Benoit et al. (2013)), further elaboration is needed.

Methodology

Major Concerns

1. Forward and Backward Directions (Lines 159–185)

o In Figure 1, Path 1 (DN) appears to be a ‘recall’ task with no alternatives, while ND is a ‘recognition’ task with six-alternative forced choices. Are these forward (DN) and backward (ND) directions truly equivalent?

o A more comparable approach might be:

DN: Seeing the number “three” briefly and hearing six audio alternatives sequentially, then selecting the correct one.

ND: Hearing the word "three" briefly and seeing six visual alternatives sequentially, then selecting the correct one.

Could this be a limitation of the design?

Minor Concerns

2. Stimuli Presentation (Lines 143–146)

The manuscript states that both laptops and printed versions were used. If printed versions are more engaging for children, this could introduce performance differences.

o Which version was used for which task? Were they switched during the experiment or research?

o How many participants used only laptops vs. only printed versions?

3. Handling of Participants Scoring Zero in the Practice Phase (Lines 197–198)

o If children who scored zero still understood the instructions, this should be explicitly stated.

o Including data from participants who did not understand the task may not be appropriate.

Results and Discussion

Major Concerns

1. Excessive Use of Statistical Tests and Lack of p-Value Correction

o Given the number of statistical tests, p-value corrections (e.g., Bonferroni) should be considered to minimize Type I errors.

o If the study follows previous research in not applying p-value corrections, this should be clearly stated. However, without corrections, the increased risk of Type I errors could lead to misleading interpretations.

2. Conflicting Arguments (Lines 224–234)

o The claim that alternative forced-choice designs influence results is not supported, as no significant differences were observed in any path in the current study, which is the argument that selecting from six alternatives (NA) is equivalent to determining the number in an array (AN).

3. Indirect Comparisons (Lines 265–271)

o The argument that one condition is below chance level, while another is above chance level, therefore one is superior is weak. A direct comparison is necessary to support this claim, especially since it involves ethnicity.

4. Chance Level Analysis (Lines 259-264, 281–287)

o The assumption that the expected chance level is 1/6 may not be appropriate for all conditions. Could this be reconsidered, particularly for conditions involving number words as outputs?

o Excluding zero scores may be problematic, as zero values still carry meaning in random selection probabilities.

5. Counting and Mapping Performance (Line 387-388)

o In Figure 3, four out of eight conditions (or three out of four in large numbers) showed significant biases, indicating partial improvement in mapping performance rather than a robust effect.

6. Potential Alternative Explanations and Counterarguments

o The study suggests that reminding children to count (using a single hand) improves their performance. However, alternative explanations should be discussed.

o For example, one key factor may be working memory. Since performance improves with age even in the absence of explicit counting (Lines 242–250), the explicit counting alone may not be the sole driving factor. Instead, working memory development could facilitate mental counting (arithmetic) without external aids. In this case, children may already does engage in counting (mental arithmetic), and explicitly counting with a single hand—potentially serving as a visible extension of working memory—could further enhance performance.

o If a full discussion is not feasible, reconsider at least acknowledging alternative explanations in the limitations section.

Minor Concerns

7. Performance Collapsing Across Directions

o Did the study collapse performance across the two mapping directions for analyses other than investigating mapping directions? This seems likely, given that Table 1 shows no differences, but this should be explicitly stated.

8. Figures and Tables

o Revise the figures and tables to ensure they align with the statistical methods used. Specifically, check whether the current study compares means between groups (Student t-test), compares distributions of independent groups (Mann–Whitney U test), or tests whether the median of paired differences significantly differs from zero (Wilcoxon signed-rank test). For the latter two methods, reconsider presenting statistics other than means, such as medians and interquartile ranges.

9. Clarification of Graphs and Tables

o Each graph and table should explicitly define terms such as "Path 1, 2, 3," "first/second direction," asterisk criteria, and abbreviations. Specifically, what do "first direction (M1)" and "second direction (M2)" represent? For example, does M1 correspond to DN and M2 to ND?

10. Statistical Results (Line 249)

o The Mann-Whitney U test results appear to be missing. Are all results significant at p < .05, .01, or .001?

11. p-Value Formatting

o In tables, p-values should not be reported as 0.000. Instead, the standard convention would be to report < .001.

12. Figure 4 Layout

o Graphs in Figure 4 should match the statistical analysis. If the focus is on the developmental order of mapping performance, then the graphs should be arranged accordingly for small and large numbers with appropriate asterisks.

Other Concerns

Major Concerns

• Data Availability

o The manuscript states that "all data are fully available without restriction", yet another section states that data is available upon reasonable request. This inconsistency should be clarified.

Minor Concerns

• Citation formatting (e.g., Benoit et al. (2013)) may need revision.

Conclusion

The conclusions drawn in this manuscript may be premature, as statistical and methodological refinements are necessary. In addition to the previously mentioned concerns, one key issue pertains to Experiment 2, where the instruction "Please count using your hand." (Line 352) is meant to remind children to use a single hand for counting, rather than simply reminding them to count (e.g., mental arithmetic). Specifically, the role of hand usage in the methodological procedure is inconsistently addressed throughout the article, including in the conclusions. Given its potential key role, generalizing the counting effect may not be appropriate without further clarification.

Additionally, please ensure that children were consistently using only one hand for counting. For instance, how did they count the number six? Was there any verification of whether they used one hand or switched to two hands? Clarifying this methodological detail would strengthen the study’s validity and interpretation of results.

Limitations and Future Directions

As noted throughout, the limitations section should explicitly acknowledge additional factors that may have influenced the results.

Reviewer #5: Your research is relevant in the field, and we believe improvements can be made in the following areas.

The literature review is not sufficiently comprehensive. Studies from the past five years account for only 20% of the references, raising concerns that key advancements in this area may have already been established. It is recommended that more recent studies be incorporated to ensure the discussion reflects the current state of research and justifies the necessity of this study.

Lines 104–114: The review on the role of counting in numerical mapping is limited, making the rationale for conducting Experiment 2 relatively weak. The necessity of this experiment is not strongly justified. It is advised to expand the discussion on counting-related research to provide a more solid foundation for this experimental design.

There are methodological concerns regarding the experimental setup. Line 201: Participants completed tasks in the order presented, which raises the possibility that increased familiarity with the tasks could lead to improved performance in later trials or restrict certain pathways. We suggest that the task order be randomized to obtain more reliable and generalizable conclusions. Additionally, the participants in the two experiments were not the same, introducing the possibility that initial differences between the groups may have influenced the results. Specifically, after introducing counting as a variable, the source of the observed differences cannot be conclusively determined, as this was not addressed in the methodology. Controlling for initial variables would enhance the validity of the findings.

Line 270: The comparison drawn here appears somewhat arbitrary. Since the data points are not from the same temporal context, they may be influenced by differences in educational background and broader societal developments over time. It is recommended that this comparison be removed or revised to present a more rigorous interpretation.

6. PLOS authors have the option to publish the peer review history of their article (what does this mean? ). If published, this will include your full peer review and any attached files.

**Do you want your identity to be public for this peer review?** For information about this choice, including consent withdrawal, please see our Privacy Policy .

Reviewer #1: No

Reviewer #2: No

Reviewer #3: No

Reviewer #4: No

Reviewer #5: No

---

## [Author Response · Author response to Decision Letter 1]

29 Apr 2025

Dear editor and dear reviewers

Re: Manuscript ID: PONE-D-25-02198 and Title: “The development of numerical mapping in preschool children”

Thank you for your letter and for the reviewers’ insightful comments regarding our manuscript. We are deeply grateful for your careful review and thoughtful suggestions, which have substantially improved the quality of our paper.

We have uploaded the revised manuscript. All revisions in the text are marked in red highlight for easy reference. Our detailed responses to the reviewers’ comments are also provided, with changes clearly highlighted. To facilitate access to our data, we have uploaded the stimuli, raw data, and related experimental materials to the Open Science Framework at https://osf.io/rs2up/. Additionally, I have updated my information in Editorial Manager and validated my ORCID iD.

This work was supported by the National Natural Science Foundation of China (grant number 32360203) and the Major Project of Key Research Bases of Humanities and Social Sciences of the Ministry of Education (grant number 22JJD10009). The funders had no role in study design, data collection and analysis, decision to publish, or preparation of the manuscript. As requested, we have included the full name of the ethics committee in the Methods section and have attached the official stamped ethics approval document.

We would love to thank you for allowing us to resubmit a revised copy of the manuscript and we highly appreciate your time and consideration.

Sincerely.

Yun Pan.

Point-by-point responses to reviewers' comments are as follows:

Reviewer 1

1. Data Availability

As per PLOS Data Policy, all data underlying the findings should be made available without restriction, unless there are specific reasons for withholding it. The manuscript states that readers must "reach out to the corresponding author with a reasonable request" to obtain the data. Please provide a clear justification for why the data cannot be shared publicly and why restrictions on sharing exist.

Response Thank you for your precious comments and advice. To facilitate readers to obtain data, we have uploaded the stimuli, raw data and related experimental materials into the Open Science Framework at https://osf.io/rs2up/.

2. Age Grouping Justification

The manuscript presents age groupings of 2-, 3-, and 4-year-olds without providing a rationale for these specific categories. It would be beneficial to explain why these particular age groupings were chosen and how they align with relevant developmental stages to support the robustness of the study design.

Response Thank you for your precious comments and advice. To ensure our sample is comparable to that of Benoit et al. (2013), we divided the children into three age groups based on the sample characteristics reported in their study. Additionally, following the presentation of average age in Benoit et al. (2013), we have revised the way we report average age accordingly, as indicated in red in the manuscript (Page 10, Lines 209-213).

Revised Text in Manuscript: “To make our sample as comparable to that of Benoit et al. (2013) we further divided the children into three age groups : 40 two-year-olds (20 male; range = 24–35 months; M = 2 years 7 months), 45 three-year-olds (23 male; range = 36–47 months; M = 3 years 7 months), and 37 four-year-olds (20 male; range = 48–59 months; M = 4 years 4 months).”

3. Sampling Strategy

While the geographic locations of participant recruitment are mentioned, the manuscript lacks a description of the sampling strategy. It remains unclear whether this was a convenience sample or if any selection criteria were applied. Please provide more detail on the recruitment process, including whether participants were randomly selected or chosen based on particular inclusion/exclusion criteria. This additional information will help assess potential biases in the sampling method and improve the transparency of the study design.

Response Thank you for your precious comments and advice. In response, we have revised the manuscript to provide a clearer description of the participant recruitment process. Specifically, we now state that a stratified random sampling method was used within each preschool to ensure both representativeness and gender balance among participants. This revision appears in the Methods section (Pages 10-11, Lines 215-216).

Revised Text in Manuscript: “A stratified random sampling method was employed within each preschool, ensuring equal representation of boys and girls across the selected age groups.”

4. Control Conditions

In Experiment 1, the manuscript does not specify what control conditions were implemented to ensure the results were not influenced by confounding variables. A clearer description of the control measures would improve the interpretability of the results.

Response Thank you very much for your helpful comment regarding the lack of detail on control conditions in Experiment 1. In response, we have added a paragraph (see Page 14, Lines 291-300) at the end of the Procedure section to clearly describe the measures taken to control for potential confounding variables. Specifically, we now include information on stimulus standardization, Latin square design for target number sequences, randomization of response option positions, consistent use of verbal instructions, experimenter training, and the use of a structured practice phase. These measures were implemented to improve the interpretability of the results. We hope this revision addresses your concern.

Revised Text in Manuscript: “To control for potential confounding variables and enhance the internal validity of the experiment, several measures were implemented. All visual stimuli—including digits and dot arrays—were standardized in font type, size, color, and spatial layout across trials to eliminate perceptual biases. The sequence of target numbers was fixed based on the first two lines of a Latin square design to reduce order effects. In the tasks involving six alternative response options, the position of the alternatives was randomly assigned to prevent positional bias. Additionally, a standardized procedure was followed, with all participants receiving identical verbal instructions administered by a trained experimenter. A structured practice phase with feedback was also included to ensure participants understood the task requirements before proceeding to the formal testing phase.”

5. Participant Selection Rationale:

While the authors state that they aimed to match the socioeconomic status of participants in Experiment 2 to those in Experiment 1, no details are provided on how this was assessed or confirmed. The different geographical location (Liangshan Prefecture vs. Dangwu Town and Mengyang Town in Experiment 1) raises questions about potential regional differences.

Response Thank you for your precious comments and advice. In response to your concern about the socioeconomic status of participants, we have provided additional details in the manuscript regarding how we ensured comparability between Experiments 1 and 2. The specific information regarding the preschools, tuition fees, and parental occupations can be found in the revised version of the manuscript (Page 21, Lines 454-459).

Revised Text in Manuscript: “All participating preschools were inclusive, non-profit institutions, with standardized tuition fees set by local government regulations, ranging from CNY 1,600 to 2,000 per semester. These schools served children from local families and did not apply any selective admission policies. Informal information obtained at the school level further suggested that most parents shared similar occupational backgrounds, primarily in small business, service industries, or skilled labor.”

6. Time Gap Between Experiments:

Experiment 1 was conducted from April to May 2023, while Experiment 2 was conducted from December 2023 to January 2024. The potential impact of this 7–8-month gap is not addressed in the methodology. Please consider discussing whether this gap might have introduced seasonal or contextual changes that could affect participants’ behaviors or experiences. If the gap is not considered impactful, a justification for this assumption would strengthen the methodology.

Response Thank you for your insightful comment. We acknowledge that Experiments 1 and 2 were conducted in different months, with a time gap of 7–8 months between them. However, both experiments involved children of similar ages and targeted foundational numerical cognition, which is generally stable and less susceptible to seasonal or environmental changes (e.g., Dehaene, 1992). Given these considerations, we believe that the time gap between the two experiments is unlikely to have introduced any significant bias. We have updated the manuscript to clarify this point (Page 22, Lines 464-469).

Revised Text in Manuscript: “The experiment was conducted from December 11th, 2023, to January 4th, 2024, while Experiment 1 was carried out from April to May 2023. Although the two experiments were conducted in different months, both involved children of similar ages and focused on foundational numerical cognition [2], which is generally stable and less influenced by seasonal or environmental changes. Given this, we believe the time gap between the two experiments is unlikely to have introduced any significant bias.”

2. Dehaene, S. (1992). Varieties of numerical abilities. Cognition, 44(1-2), 1-42.

7. Sample Representation and Generalizability:

The participants were recruited from five preschools in two cities in China. While this is a reasonable geographical spread, it would be useful to comment on the demographic characteristics of the children. This would help assess the generalizability of the findings beyond this specific sample. The homogeneity of the sample should also be discussed to evaluate the external validity of the study's conclusions.

Response Thank you for your insightful comment. We agree that providing more details on the demographic characteristics of the participants is important for evaluating the generalizability of our findings. The children in our study were recruited from five preschools in two cities in China, and the sample was relatively homogeneous in terms of socioeconomic status, as all participating preschools were inclusive, non-profit institutions with similar tuition fees. Informal information from the schools indicated that most parents worked in small businesses, service industries, or skilled labor.

First, we have already expanded the demographic information in the Methods section in Experiment 2 (Page 21, Lines 454-459) while answering the fifth question. Second, we have addressed these points in the Limitations section (Page 32, Lines 700-705), where we acknowledge that the sample may not fully represent children from other regions or socioeconomic backgrounds beyond this local context. We suggest that the findings may be more generalizable to similar populations within China and emphasize that further research with more diverse samples would be valuable for assessing the broader applicability of the results.

Revised Text in Manuscript: “First, the participants were recruited from three towns across two provinces, which may limit the generalizability of the findings. The study also employed a cross-sectional design with a fixed task order, potentially introducing order effects and individual variability. Future research should consider broader sampling, longitudinal tracking, and fully randomized within-subject designs to enhance generalizability and experimental control.”

8. Reminding Children to Count:

The inclusion of the "reminding children to count" condition in Experiment 2 is an interesting and relevant manipulation. However, more details on how the counting reminder was delivered separately and its duration would be helpful for evaluating the effectiveness of this intervention. Was there a standardized method for reminding the children to count, and how was it ensured that the reminder was consistent across participants?

Response Thank you for your insightful comment. In response, we have added further details to the Procedure section of Experiment 2 (see Page 22, Lines 471-477) to clarify how the counting reminder was delivered. Specifically, the reminder was standardized across all dot-related tasks and participants: the experimenter instructed each child to use their right index finger to point at the dots one by one while verbally counting them before responding. This consistent procedure ensured that the reminder was delivered uniformly across sessions and participants.

Revised Text in Manuscript: “Based on Experiment 1, Experiment 2 introduced a condition to remind children to count. In the AN, NA, AD, and DA tasks, the experimenter gave a standardized instruction before the child responded, saying, “Please count using your hand. How many dots are there?” To ensure consistency across participants, the experimenter instructed each child to use their right index finger to point at the dots one by one while verbally counting them aloud. This procedure was strictly followed to standardize the counting reminder across all children and trials.”

Reviewer 2

1. Introduction Section. The current introduction lacks sufficient elaboration on real-world problems and research background.

Response Thank you for your valuable feedback. In the revised manuscript, we have significantly reorganized and expanded the Introduction section to enhance its clarity and logical flow. Specifically, we have added clear subheadings. These changes help to better elaborate on the real-world problems, theoretical background, and research motivations relevant to our study. We believe that the revised structure provides a more comprehensive and coherent context for readers and addresses the concern raised. We sincerely hope that the improvements meet your expectations.

2. Literature Review. It is recommended to add a dedicated "Literature Review" section. Reorganize existing content from the Introduction to systematically summarize prior studies with logical taxonomies.

Response Thank you for your helpful suggestion. In the revised manuscript, we have created a dedicated Literature Review section and reorganized the relevant content that was previously embedded in the Introduction, systematically summarizing prior research with a clearer logical structure. Specifically, we have added clear subheadings, including Mapping Direction, Developmental Order of Numerical Mapping, Mapping Differences by Number Size, Effect of Counting on Numerical Mapping, and The Current Study. We believe these changes enhance the clarity, coherence, and accessibility of the background information. We appreciate your guidance in helping us improve the manuscript.

3. Reference Timeliness. Add some relevant studies within five years.

Response Thank you for your valuable suggestion. In the revised manuscript, we have carefully reviewed and incorporated additional recent literature. Specifically, we added 20 important references published within the past five years. As a result, the manuscript now includes 31 references from the past five years out of a total of 71 references, accounting for 43.66%. We believe that these updates enrich the manuscript and ensure that it reflects the latest developments in the field. We sincerely appreciate your feedback, which has helped us strengthen the quality and relevance of the paper.

4. Comparative Analysis of Experiments. The comparative analysis between the two experiments is insufficient.

Response Thank you for your insightful comment. In the revised manuscript, we have strengthened the comparative analysis between Experiment 1 and Experiment 2. Specifically, Experiment 2 was designed based on Experiment 1 by adding a “counting reminder” condition, aiming to further explore how reminding children to count influences the evaluation of the developmental order and the symmetry of mapping direction in numerical mappings.

In the Results and Discussion and General Discussion sections, we systematically compared the two experiments from three perspectives: (1) the symmetry of mapping directions, (2) the developmental level of numerical mappin

---

## [Decision Letter · Decision Letter 1]

The development of numerical mapping in preschool children

PONE-D-25-02198R1

Dear Dr. Pan,

We’re pleased to inform you that your manuscript has been judged scientifically suitable for publication and will be formally accepted for publication once it meets all outstanding technical requirements.

Kind regards,

Alessandro Bruno, Ph.D.

Academic Editor

PLOS ONE

Additional Editor Comments (optional):

Dear Authors,

Thanks for your hard work of revision.

I recommend your paper for acceptance.

my best regards,

A.B.

Reviewers' comments:

Reviewer's Responses to Questions

Comments to the Author

1. If the authors have adequately addressed your comments raised in a previous round of review and you feel that this manuscript is now acceptable for publication, you may indicate that here to bypass the “Comments to the Author” section, enter your conflict of interest statement in the “Confidential to Editor” section, and submit your "Accept" recommendation.

Reviewer #1: All comments have been addressed

Reviewer #2: All comments have been addressed

2. Is the manuscript technically sound, and do the data support the conclusions?

Reviewer #1: Yes

Reviewer #2: Yes

3. Has the statistical analysis been performed appropriately and rigorously? 

Reviewer #1: Yes

Reviewer #2: Yes

4. Have the authors made all data underlying the findings in their manuscript fully available?

Reviewer #1: Yes

Reviewer #2: Yes

5. Is the manuscript presented in an intelligible fashion and written in standard English?

Reviewer #1: Yes

Reviewer #2: Yes

6. Review Comments to the Author

Reviewer #1: Thank you for submitting the revised manuscript. It carefully addresses the previous concerns, effectively incorporates the comments, and is suitable for publication.

Reviewer #2: (No Response)

7. PLOS authors have the option to publish the peer review history of their article (what does this mean? ). If published, this will include your full peer review and any attached files.

**Do you want your identity to be public for this peer review?** For information about this choice, including consent withdrawal, please see our Privacy Policy .

Reviewer #1: No

Reviewer #2: No

---

## [Editor Report · Acceptance letter]

PONE-D-25-02198R1

PLOS ONE

Dear Dr. Pan,

I'm pleased to inform you that your manuscript has been deemed suitable for publication in PLOS ONE. Congratulations! Your manuscript is now being handed over to our production team.

Kind regards,

on behalf of

Associate Professor Alessandro Bruno

Academic Editor

PLOS ONE